# Benchmarking Flexible Electric Loads Scheduling Algorithms

**Koos van der Linden, Natalia Romero and Mathijs M. de Weerdt \***

Faculty of Electrical Engineering, Mathematics and Computer Sciences, Delft University of Technology, Van Mourik Broekmanweg 6, 2628 XE Delft, The Netherlands; J.G.M.vanderLinden@tudelft.nl (K.v.d.L.); n.romero.lane@gmail.com (N.R.)

**\*** Correspondence: m.m.deweerdt@tudelft.nl

**Abstract:** Due to increasing numbers of intermittent and distributed generators in power systems, there is an increasing need for demand responses to maintain the balance between electricity generation and use at all times. For example, the electrification of transportation significantly adds to the amount of flexible electricity demand. Several methods have been developed to schedule such flexible energy consumption. However, an objective way of comparing these methods is lacking, especially when decisions are made based on incomplete information which is repeatedly updated. This paper presents a new benchmarking framework designed to bridge this gap. Surveys that classify flexibility planning algorithms were an input to define this benchmarking standard. The benchmarking framework can be used for different objectives and under diverse conditions faced by electricity production stakeholders interested in flexibility scheduling algorithms. Our contribution was implemented in a software toolbox providing a simulation environment that captures the evolution of look-ahead information, which enables comparing online planning and scheduling algorithms. This toolbox includes seven planning algorithms. This paper includes two case studies measuring the performances of these algorithms under uncertain market conditions. These case studies illustrate the importance of online decision making, the influence of data quality on the performance of the algorithms, the benefit of using robust and stochastic programming approaches, and the necessity of trustworthy benchmarking.

**Keywords:** flexibility; energy markets; benchmarking; online optimization; simulation





## 1. Introduction

The integration of renewable energy is central for achieving energy security in a zero-carbon energy future [1]. Furthermore, electrifying a significant part of transportation brings further economic and environmental benefits. Most renewable energy sources are intermittent. Thus, maximizing their use requires different operation and planning strategies to those traditionally used for controllable generators. Exploiting the flexibility in demand, in particular, by flexibly charging batteries of electric vehicles, is a viable strategy for coping with the additional uncertainty [2].

The scientific community has been responsive to this by developing a broad spectrum of algorithms with which to design effective incentive programs, commonly known as demand response (DR) programs. Over 70 publications of demand-side management were reviewed in [3] to establish a general framework for such approaches; the authors analyzed whether users made selfish or cooperative decisions; the problem is solved with deterministic or stochastic methods, and the algorithms are offline versus online. Deng et al. summarized the objectives and issues in DR programs [4], and Mukherjee and Gupta the control type considered in smart scheduling algorithms [5]. One of the most extensive categorizations of DR programs and algorithms was published in [6]; it accounts for over 200 publications.

With so many options, power sector stakeholders need tools to compare methods and identify the method that suits their objectives the best. Unfortunately, only a small number

of publications focus on benchmarking existing methods or comparing new contributions to established ones. In [7], a new approach that uses hierarchical control was compared to algorithms assuming central control, which are known for finding system-wide optimal solutions. Four models to maximize the value of flexible resources were proposed and compared in [8].

The challenge of assessing strategies increases when stakeholders consider real-time decisions, which require online algorithms that update decisions based on new information. Simulations allow evaluating the performances of such online algorithms based on multiple criteria. Furthermore, simulations can help with developing better online optimization algorithms for complex dynamic problems [9].

### 1.1. Related Work

A broad spectrum of tools and simulators have been developed to address the upcoming challenges in the power sector. Conolly et al. reviewed computer tools that supported decisions relevant to the integration of renewable energy [10]. This survey aimed to inform decision-makers about the best tool to use for their specific situations. Tools were classified into categories: simulation, scenario, equilibrium-tool, top-down, bottom-up, operation optimization, and investment. None of these classifications focuses on assessing the strategies or algorithms used for scheduling loads.

Soares et al. classified simulation tools in the electricity market, and micro-grid and smart grid simulators [11]. Energy market simulators model the energy market as a series of interactions between different stakeholders negotiating electricity contracts. Some examples are: the Simulator for Electric Power Industry Agents (SEPIA) [12], Powerweb, which includes both a market model and a grid model [13], the Short-medium Run Electricity Market Simulator (SREMS) based on game theory [14], the Electricity Markets Complex Adaptive System (EMCAS) [15], the AMES, a wholesale power market test bed [16], the advanced energy systems analysis model, EnergyPLAN [17], and the Multi-agent Simulator of Competitive Electricity Markets (MASCEM) [18]. Many of these energy market simulators are very powerful and include agent models that can learn from their actions. For example, ALBids is a decision support tool for agents negotiating in the energy market [19]. It includes different strategies already introduced in the literature. These strategies can be used by agents in MASCEM and compared considering different market environments.

Seventy-five modeling tools to analyze energy systems were studied by Ringkjøb et al. [20]. Similarly to [10], this more recent study highlights that energy computer tools are designed to answer specific questions. Among the tools included, the Distributed Energy Resources Customer Adoption Model Plus (DER-CAM+) has an objective relevance to our work [21]. This tool classifies loads based on their fuel or end-use type. It integrates a power grid model with market data to represent energy and reserve markets. Selecting the best strategy for distributed loads considering the market participation and grid constraints is one of its main objectives.

The tools studied in these surveys have many applications, but they are mostly based on deterministic models [20]. Furthermore, when agents can consider alternative strategies, these are predetermined or derived from the learning algorithms dependent on the market simulation [22]. Of the tools studied, ALBids and DER-CAM+ aim to find an optimal strategy for parties with distributed and flexible loads participating in energy markets. However, these tools include this capability among an array of options, which hinders their use for studying the effects of different factors (under controlled settings) on algorithms for scheduling decisions. What is missing in the literature is an objective way to evaluate different algorithms that support scheduling and trading flexible loads in multiple electricity markets in an online context where more information becomes available over time.

### 1.2. Contributions

This work presents a conceptual framework with which to benchmark offline and online algorithms for scheduling flexible energy loads, for instance, for charging the batteries of electric vehicles (EVs). This framework is implemented by a simulator and is available together with implementations of a number of state-of-the art algorithms, open-source, in a toolbox called B-FELSA [23]. B-FELSA addresses the absence of a comparison methodology and simulation environment for algorithms that optimally schedule flexible loads. The toolbox includes a simulator which allows for comparing different algorithms, checking for reproducibility of observed behavior, and performing sensitivity analysis to input parameters. Specifically, it allows for comparing algorithms in an uncertain environment that changes over time by providing repeatedly updated look-ahead information.

This benchmarking framework was designed from the perspective of a consumer with flexible demands (or a stakeholder acting on his/her behalf) who wants to minimize total costs considering market incentives for load shifting. It covers shiftable and energy-based electric devices and accounts for the uncertainty associated with market incentives. This agent is a price-taker. The uncertainty in the different markets is simulated through scenarios of prices and settlement decisions. The toolbox has been used in one earlier publication to compare methods that schedule flexible loads for trading in electricity markets with a shared market participation constraint [24]. To illustrate the use of this framework, straightforward and advanced heuristics, deterministic optimization, and stochastic approaches were tested in a realistic scenario comparable to the Dutch energy market, and in an artificial scenario in which the quality of the data could be controlled.

In summary, the contributions of this work include: (1) the framework itself, i.e., a verification method for online methods in uncertain information environments, implemented in a simulation environment which offers tools for calibration and validation tasks, making it an effective benchmarking tool [25]; (2) representations of different market models and relevant data sources for EV charging demand and market prices; (3) implementations of seven solution methods, including simple and fast heuristics and state-of-the-art stochastic algorithms; and (4) two case studies to illustrate the potential of the framework for comparing different methods and analyzing the effect of new (better) information.

The rest of this paper is organized as follows: Section 2 describes and motivates the scheduling context of the benchmark. Section 3 describes the framework and how it addresses the challenge of benchmarking online algorithms. This includes descriptions of the planning algorithms that are evaluated. Section 4 presents two case studies and discusses the results. Section 5 concludes this article.

## 2. Load Scheduling Context

Transmission system operators (TSOs) and distribution system operators (DSOs) around the world have identified the benefits of using DR programs to prevent congestion or ensuring balance of supply and demand. This has led to large numbers of different programs, tariffs, and market designs that are employed to elicit the flexibility in loads from mainly residential customers, commercial customers, or both. First we explain and motivate which DR context we have chosen for the benchmarking of load scheduling algorithms, and then we give the type of flexible load that is represented.

### 2.1. Demand Response Programs

As described in the surveys on demand side management and demand response [3,4,6], DR programs include centralized and decentralized control; advanced tariffs such as time of use, critical peak, real-time pricing, and tier tariffs; and short-term electricity markets.

Considering current practice, the short-term electricity and ancillary services markets seem to be the preferred way of trading flexible demand. The most common of these markets consists of a two-sided auction where parties trade energy supply and demand in hourly or block intervals called program time units (PTUs) [26], and typically start

about half a day before the day of delivery. Such an auction is known as the day-ahead (DA) market.

Some countries have market designs that allow trading between stakeholders closer to the time of delivery. These markets are known as intraday markets in European countries. They facilitate trading continuously, or sometimes in discrete slots, after closure of the DA market. A similar environment is known as the adjustments market by the Electric Reliability Council of Texas (ERCOT).

Parallel to DA and intraday markets, many TSOs use (voluntary) ancillary services to maintain grid stability. Trading parties can make bids to offer reserves supporting regulation up or down. Regulation up occurs in any instance when the demand exceeds the supply. The opposite case is known as regulation down. The deadline to place bids for offering ancillary services varies across countries, and conditions are typically different for offering voluntary or contracted services. A bid typically consists of both a volume and a price, possibly different for each PTU. Offering ancillary services is equivalent to a demand bidding and buyback DR program, but with the option of also increasing consumption.

After energy delivery, differences between actual energy use and the DA commitment are settled by paying an imbalance price to the TSO. This is called the imbalance market.

In the benchmark we consider participation in the DA, imbalance, and reserve markets. Intraday participation is not included but could be developed using the same logic used for online decisions in the imbalance market.

### 2.2. Type of Load and Its Flexibility

Electric devices in the grid can be classified in fixed, shiftable, and elastic energy-based or comfort-based [3]. Shiftable devices have a fixed load profile that can only move in time. Energy-based elastic devices must consume a fixed amount of electricity within a given window; and comfort-based elastic devices have flexible profiles only constrained by a comfort level target, e.g., temperature control systems such as heat pumps [3]. Furthermore, devices with energy storage capacity can offer unidirectional or bidirectional services (when devices can sell energy to the system) [5].

The analysis presented here is on elastic devices, as these are more flexible than shiftable devices and also present a more interesting scheduling problem. In particular, we focused on electric vehicle (EV) charging. EVs have relatively high flexibility and capacity, and this makes them good candidates for providing DR. In fact, EVs form one of the most important sources for flexibility related to transportation electrification. They behave similarly to other energy-based elastic devices such as a pool pumps or even a pumping station. Moreover, when we allow (unmet) demand violations, as some models do (four out of seven included in this work), EV charging can also represent comfort-based elastic devices such as heat pumps.

The resulting scheduling problem is computationally challenging under conditions such as a shared constraint on multiple loads with already complete information [27]. It is therefore relevant to benchmark methods not only regarding the costs/benefits and unmet demands of the resulting trading decisions, but also their computational performances (i.e., runtime of the algorithm). The most challenging, however, is to compare how the different methods deal with the uncertainty (e.g., regarding prices) and the information regarding this that becomes available over time.

### 3. Overview of B-FELSA

The previous section highlights the broadness of the scope of the DR program context. The large variety in the problem's context has resulted in a large number of solution methods. This also makes comparing and benchmarking solution methods a hard task. B-FELSA is designed to deal with this complexity.

B-FELSA has a modular design, which makes it a flexible platform for testing, comparing, and designing new algorithms in various market design settings. The modules are grouped based on the steps followed for evaluating algorithms. The first phase, i.e., the

input phase, is managed by the data provider. The data provider uses the flexible load and market environment modules to configure the input data. The next two phases and modules are critical for benchmarking algorithms. They use the market simulation environment to optimize scheduling decisions and measure the outcomes of those decisions. The modules are named solution method and evaluator, respectively. Table 1 contains an overview of the modules of B-FELSA and Figure 1 illustrates the structure of B-FELSA. We have made B-FELSA open-source, and it can be obtained at [23].

**Table 1.** An overview of the modules of B-FELSA and the included alternatives/parameters which can currently be used.

| Module | Short Description | Included Alternatives |
|---|---|---|
| Data provider | Generates the data for the evaluation and scenario data for the predictions, based on the configuration and the market model. | Historic data (from Dutch or ERCOT market); generated from market model (ARMAX). |
| Flexible load | Models the behavior of a flexible load | Charging EVs |
| Market model | Models the behavior and rules of the electricity market | Day-ahead, reserves, Dutch market rules, ERCOT market rules, V2G, variable PTU length, reserve commitment deadline |
| Solution method | Solves the problem | Heuristics; stochastic programming (one stage and two stage); deterministic solution; quantity-only. |
| Evaluator | Evaluates the solution based on the problem set-up and market model | Runtime, costs, unmet demand and exceeded battery capacity. |

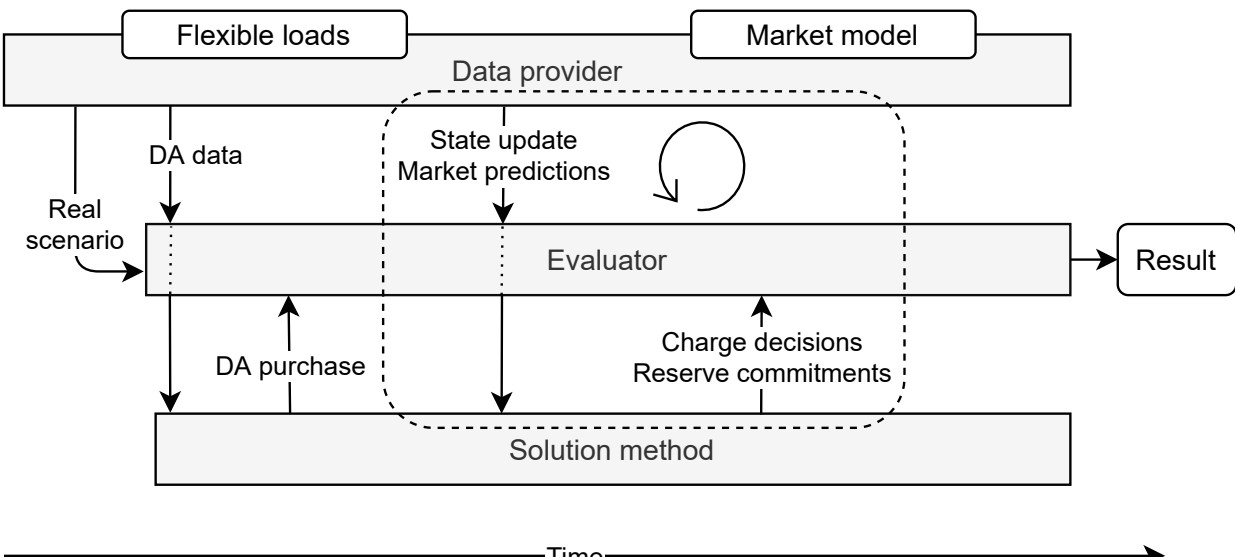

**Figure 1.** The interactive flow between the simulation and the solution method. The data provider generates a real scenario, which is then used in the simulation. The solution method makes day-ahead (DA) purchases based on DA price data. The simulation then proceeds step by step through every time step, providing the solution method with updated information from the data provider and receiving new decisions from the solution method.

Each of these modules is discussed in a separate section below.

### 3.1. Data Provider

The input data consist of the loads, market, model, and simulator configuration. The market data include DA market prices, the regulation price, and ancillary service usage. Additional data include information such as grid constraints, and parameters defined by the user to model the uncertainty and its evolution.

Every load has a session start and end, a starting state of charge (SOC), a required minimum SOC, a capacity, and a maximum energy usage (charging speed). The market data provided to the solution models are the DA price per hour, and a set of scenarios consisting of an up and down regulating price, an imbalance price, and the percentages of up and down reserve usages for every PTU. An overview of the inputs and outputs of the solution methods is included in Table 2.

**Table 2.** An overview of the inputs and outputs for the solution models. At the start, information is provided for every flexible load. Then day-ahead information is provided and day-ahead purchase decisions are made. Then in the online simulation, every time step, new scenario data are provided and new decisions are made. For explanations of unmet demand and exceeded battery capacity, see Section 3.5.

| | | Input | Output/Decisions |
|---|---|---|---|
| **Simulation start** | Per EV | Arrival Time<br>Departure time<br>Arrival SOC<br>Required final SOC<br>Battery capacity<br>Charging speed | Costs<br>Amount of unmet demand<br>Amount of exceeded battery capacity<br>Costs including penalties<br>All recorded decisions (both final and predicted) |
| **Day ahead** | Per future hour<br>Scenarios for future PTUs | DA price<br>Imbalance price<br>Up/down regulation prices and usage | DA purchase |
| **Online** | Per EV | Current SOC | Charge amount<br>Up/down reserves bid (both volume and price) |
| | Scenarios for future PTUs | Imbalance price<br>Up/down regulation prices and usage | |

### 3.2. Flexible Load

The flexible load module is designed for EV charging, but with the option of extending it with other energy-based elastic or time-shiftable devices. The algorithms included in B-FELSA consider a single type of flexible load that can be aggregated. The user can implement changes to B-FELSA to account for a mixed pool of flexible loads, as explained in [7], or to consider micro-grid cost minimization or utility and maximization problems.

### 3.3. Market Model

As DA market prices are relatively easy to predict, they are modeled deterministically in the benchmark, but the uncertainties associated with the final regulation condition (i.e., whether ancillary services are accepted) and real-time energy prices are modeled through scenarios.

The simulation environment consists of a series of discrete trading times from the DA to delivery time. With each time step the information is updated. All decisions made for energy delivery in previous steps are fixed, but all other decisions can be updated. All decisions must be aggregated to verify that the total energy demand is fulfilled within the load constraints.

The user can generate scenarios to simulate the decision environment using different methods. One of the models included to generate market price scenarios is the auto-regressive moving average model with exogenous variables (ARMAX), which is widely used for scenario generation and DA market price forecasting [28–32]. In particular, we adapted the method used by Olsson and Soder [29] for real-time balancing market price

generation. This method was identified as one of the best performing by Klaeboe [31]. Both [29,31] focus on the Nordic power market prices. Olsson's method was adapted by using a Box-Cox transformation [33] instead of a log transformation to normalize the data, and by using exogenous variables to capture the seasonal (daily) component [34].

Ancillary services are used when the supply and demand need to be balanced. Acceptance happens on a continuous basis, in contrast to the discrete program time units (PTUs) for offering the service. Furthermore, the required volume also determines acceptance. Therefore, we use an abstraction that represents these two conditions. For each PTU, we estimate the percentage of time that the total energy offered is completely accepted and deployed, as was similarly done in [35]. Scenarios for this abstraction are also generated. We do this using a discrete Markov model with transition probabilities depending on the time of the day and season of the year. The transition probabilities are based on historic data.

Figure 2 shows how the ARMAX model is used to generate both an evaluation scenario (the real scenario) and price forecasts. During online evaluation, price forecasts are regenerated at every time step, creating a series starting at the current time in the evaluation. Multiple scenarios are generated using Monte Carlo simulation. Scenarios are also generated for the ancillary service usage using the Markov model introduced above. This models the market uncertainty and the increase in information as time draws closer to time of delivery.

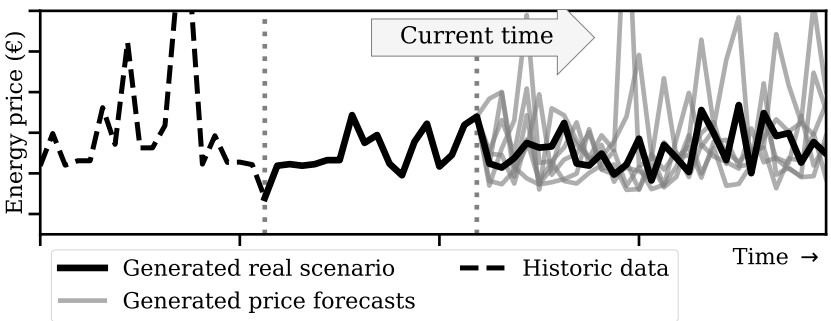

**Figure 2.** The auto-regressive moving average model with exogenous variables (ARMAX) is used to generate a real scenario (for evaluation) from historic data. ARMAX is then used to generate a number of price forecasts from the scenario. These price forecasts are updated at every time step, starting from the current time (second vertical line in the figure).

### 3.4. Solution Methods

An algorithm needs to decide the amount of energy purchased on the DA market per hour, the energy use by the flexible load, the amount of up and down reserves committed, and the up and down reserve price bids per PTU. The simulation then proceeds step by step through every time step, providing the solution method with updated information from the data provider and receiving new decisions from the solution method as illustrated in Figure 1. For a mathematical formulation of the complete problem, see [36].

Solution methods include linear and mixed integer programming, non-linear programming, dynamic programming, particle swarm optimization, Markov process based-methods, evolutionary algorithms, and other meta-heuristics [5,6]. Additionally, game theory models can be used when considering solutions that rely on the stakeholders' interactions and their willingness to cooperate.

B-FELSA can evaluate all such algorithms except for multi-party cooperating and competition models. The following are already included in the toolbox:

1.  The direct model (DI) represents buying energy immediately when plugged in.
2.  The optimal price method (OP) minimizes the costs by charging at those times when the predicted energy market price is minimal. It does not provide reserves.
3.  Heuristics can be used to provide reasonable charging decisions. The MaxReg heuristic method (MR) from [37] is included. MR follows a preferred operating point (POP).

With MR the POP is defined in such a way that it allows for maximum reserve participation. Reserve bids are quantity-only. For the analysis in this paper, the method has been updated to consider the reserve commitment deadline. It assumes that all its bids will be accepted and fully deployed and makes new robust bids based on this assumption.

4. With the deterministic model (DT), a user can plan energy and reserves market participation. The user determines a desired acceptance probability, which is used to find the optimal quantity and price bid for bidding in the reserves market. It also optimizes DA, and imbalance markets participation. The algorithm is the interpretation by Van der Linden et al. [36] of the solution method introduced by Sarker et al. [38].

5. The quantity-only method (QO) offers ancillary services but does not provide a price bid and assumes to be always accepted. It is implemented with DT by setting the desired acceptance probability to 100%.

6. The stochastic optimization model uses a number of price scenarios to determine optimal reserve price bids and energy trade in DA and imbalance markets. Two versions are included:

   (a) A one stage stochastic optimization model (SO1) that is similar to the deterministic model presented above, but optimizes for multiple scenarios instead of only the average scenario.

   (b) A two stage stochastic optimization model (SO2) that uses binary variables to determine whether a price bid will be accepted or not. It was originally developed by Van der Linden et al. [36] and improved by making the MIP formulation more tight and compact, as presented in the work where this model is used for dealing with market participation constraints [24].

B-FELSA also contains a perfect information (PI) solution model. Its solutions are obtained by providing SO2 with only the real scenario as input. The solution from PI can be used to measure how close methods perform compared to optimal decisions with perfect foresight.

All of the above methods' aim to solve the same problem and are evaluated in the same way. However, the way in which the problem is modeled may differ across these methods. For example, the deterministic model ignores any uncertainty by using only the expected future prices; the quantity-only method simplifies the decisions made by fixing the acceptance probability. Both single and two-stage stochastic optimization methods are included, and only in the two-stage model is the bid acceptance explicitly represented. B-FELSA facilitates comparing such methods that use different mathematical models under uncertain conditions in a fair and consistent way.

### 3.5. Evaluator

The evaluator uses simulations to measure the realization of decisions made in every time step. Therefore, this module is critical to assess how online algorithms use new information to improve results.

The evaluator measures the runtime and operation costs of the algorithm. Due to the uncertain usage of reserves, the evaluator also should measure the unmet demand and the exceeded battery capacity. It is possible that the algorithm makes a reserve commitment but the EV is not able to fulfill this commitment, because its battery capacity is reached. In this case the simulation continues as if it were possible, and the battery overflow is measured and output at the end. The same happens when the consumer's demand is not met.

### 4. Case Studies

The main purpose of B-FELSA is to be able to compare different online solution methods for DR programs quantitatively. In this section B-FELSA is used to study two case studies. The source code of all solution methods and the data for the case studies are available at [23]. The case studies are used to answer the following questions:

1. How does an online algorithm perform in comparison to the optimal decision with perfect information?
2. What algorithm offers the best trade-off between scalability and solution quality (feasible running time for minimum cost and risk of unmet demand)?
3. What market participation strategy offers the minimum energy costs within the desired level of risk?
4. How does the algorithm deal with new (better) information?

The first case study was a realistic case study with a market configuration similar to the Dutch energy market. It tested the performances of the algorithms mentioned in terms of operational costs, runtime, and ability to deal with uncertainty.

The second case study was an artificial case study in which the (increase of the) prediction quality of the market realization could be controlled. It measured the effect of the (increase of) prediction quality (over time) on the algorithm's performance.

All the solution methods were coded in Java. Gurobi 8.1.0 [39] was used as an MIP solver for QO, DT, SO1, and SO2, with the MIP gap being set to 1%. Runtime results are for an Intel i7 6600U CPU with 8 GB of RAM.

### 4.1. Dutch Energy Market Case Study

TenneT's market data [40] were used to create 95 historic scenarios. The ARMAX and Markov model were used to generate 10 scenarios per historic scenario, giving 950 of these scenarios in total. Then, at every increment of 15 min in every run, a total of 25 scenarios was generated using the same ARMAX and Markov model to represent forecasts. This (renewed) data were given to the algorithms to make or update their decisions. The overnight charging of one EV was simulated for each scenario. The charging session took 12 h, so 48 time steps of 15 min. The EV had a battery capacity of 30 kWh, an initial SOC of 1 kWh, needed 26 kWh, had a maximum charging speed of 7 kW, and had a charging efficiency of 90%. Unmet demand was penalized with €60/MWh and battery capacity overflow with €200/MWh (in comparison, the average DA energy price was €32/MWh). Discharging, or vehicle-to-grid, was not allowed.

The Dutch energy market has DA prices per hour, and regulation and imbalance prices per PTU of 15 min. In this case study only voluntary secondary reserve bids are considered and these regulation bids are asymmetric and can be made up of seven PTUs before delivery. Regulation is rewarded based on the amount of reserves deployed. Imbalance prices are based on the highest (lowest) price of the deployed reserve bids. The Dutch market has a minimum regulation bid size, but this is ignored in this experiment.

Based on initial exploratory experiments to establish which values work best, the desired acceptance probability for DT is set to 50%, and to 80% for SO1. SO2 optimizes based on 20 scenarios, sampled from the 25 generated scenarios.

Table 3 summarizes the results of this case study. The values shown in the table are averages across all 950 scenarios. The unmet demand and exceeded battery capacity are reported as percentages of the requested load and battery capacity respectively. An average unmet demand of 1.6% for example (the average unmet demand for DT) means that the battery is charged to 26.6 kWh, 0.4 kWh below its requested amount of 27 kWh. With a charging speed of 7 kW, this means that the EV missed approximately 4 min of charging time. This unmet demand is reflected in the cost with the penalization of €60/MWh, resulting in a penalty of 2.5 cents. A surprising result might be the "costs" with the perfect information (PI) method. In practice, it is impossible to know future market results, but this benchmark method shows that with perfect future information it is indeed possible to make a profit, so negative costs.

**Table 3.** Results for the Dutch case study. The values shown are the means $\pm$ the standard deviations of the results. The results are averages for one charging session taken across 950 scenarios.

|  | Costs + Penalty (€) | Unmet Demand (%) | Exceeded Capacity (%) | Run Time (s) |
|---|---|---|---|---|
| DI | $0.47 \pm 0.51$ | 0.0 | 0.0 | $1 \times 10^{-3} \pm 2 \times 10^{-3}$ |
| OP | $0.39 \pm 0.44$ | 0.0 | 0.0 | $1 \times 10^{-3} \pm 1 \times 10^{-3}$ |
| MR | $0.27 \pm 0.46$ | 0.0 | 0.0 | $1 \times 10^{-3} \pm 6 \times 10^{-3}$ |
| QO | $0.28 \pm 0.50$ | $0.08 \pm 0.68$ | $0.22 \pm 0.80$ | $0.59 \pm 0.10$ |
| DT | $0.21 \pm 0.54$ | $1.63 \pm 2.84$ | $0.33 \pm 1.51$ | $0.58 \pm 0.08$ |
| SO1 | $0.27 \pm 0.48$ | $0.11 \pm 0.69$ | $0.02 \pm 0.21$ | $0.66 \pm 0.10$ |
| SO2 | $0.19 \pm 0.58$ | $0.24 \pm 1.14$ | $0.17 \pm 1.03$ | $73.8 \pm 41.2$ |
| PI | $-0.25 \pm 0.78$ | 0.0 | 0.0 |  |

The results for operation costs show relatively small differences between the methods. A student's t-test, however, proves that these differences are significant with $p < 5\%$, except for MR, QO, and SO1. The differences in unmet demand are significant for all cases, except for again QO and SO1. Differences in exceeded battery capacity are also significant for all cases except for QO and SO2. It can be concluded therefore that SO2 on average has the best results in terms of costs plus penalty. The difference between SO1 and DT shows that solving for multiple scenarios does not decrease operation costs, but does decrease unmet demand and exceeded battery capacity (this is also the case when SO1 is run with the desired acceptance probability set to 50%). Interestingly, MR has cost results similar to QO and SO1 without having a penalty. What makes it more interesting is that MR does not use the scenario data at all. It also has a runtime orders of magnitude smaller.

A more detailed representation of these results across all the scenarios can be found in Figure 3. This so-called quantile plot shows for a given value on the vertical axis (such as costs or unmet demand (%)) the percentage (quantile) of scenarios with a lower or equal value (on the horizontal axis). The plot in the center shows that for all methods except DT in at least 90% of the scenarios there is no unmet demand and that this is never more than 12%. Similarly, the plot on the right shows that in 90% of the cases the battery limit is not exceeded, and that this is never more than 14%. Of course, with perfect information (PI) these constraints are never violated, and also the robust and slightly conservative MaxReg (MR) method ensures this. The violation of these constraints by some of the other solution approaches comes from the uncertainty in the realization of the reserves and the fact that they aim at minimizing (expected) cost without guaranteeing non-violation of these constraints.

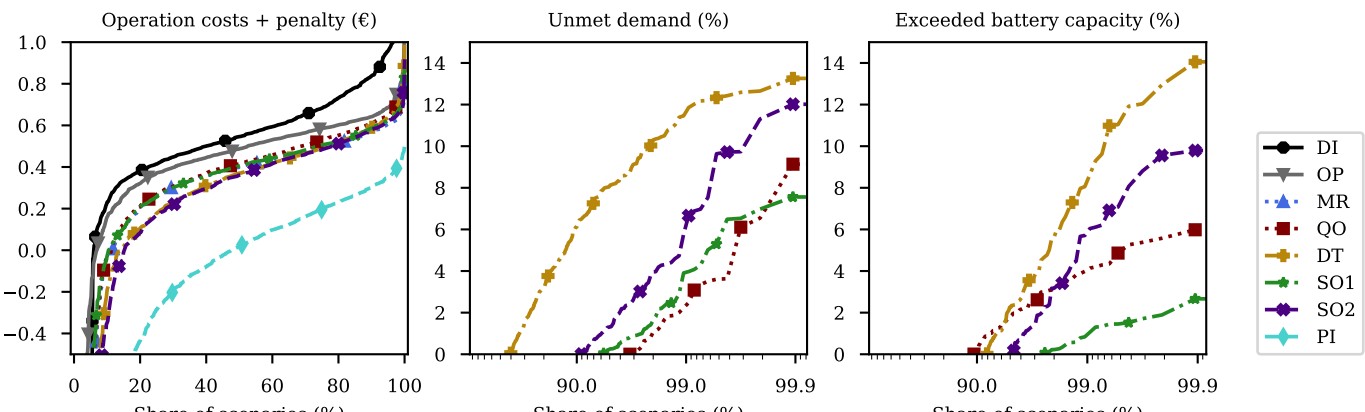

**Figure 3.** Results for the Dutch case study. The results show the performance for the $x$-th quantile in terms of operation costs plus penalty, unmet demand, and exceeded battery capacity. Note that the horizontal axes of the second and third plot are logarithmic axes.

Figure 4 shows what strategy the solution methods employ in buying/selling energy. An interesting difference is the ratio between committed and delivered reserves. The more risk-averse methods SO1 and QO commit less reserves, whereas DT and SO2 commit the largest amounts of reserves (specifically down reserves). MR is the interesting odd-one-out. It delivers a large amount of reserves without committing as much as DT or SO2. This difference can (partly) be explained by the fact that it commits quantity-only bids. The PI solution of course knows precisely when to provide reserves and has the best committed-to-delivered ratio. Interestingly, the PI solution buys most it energy in the imbalance market.

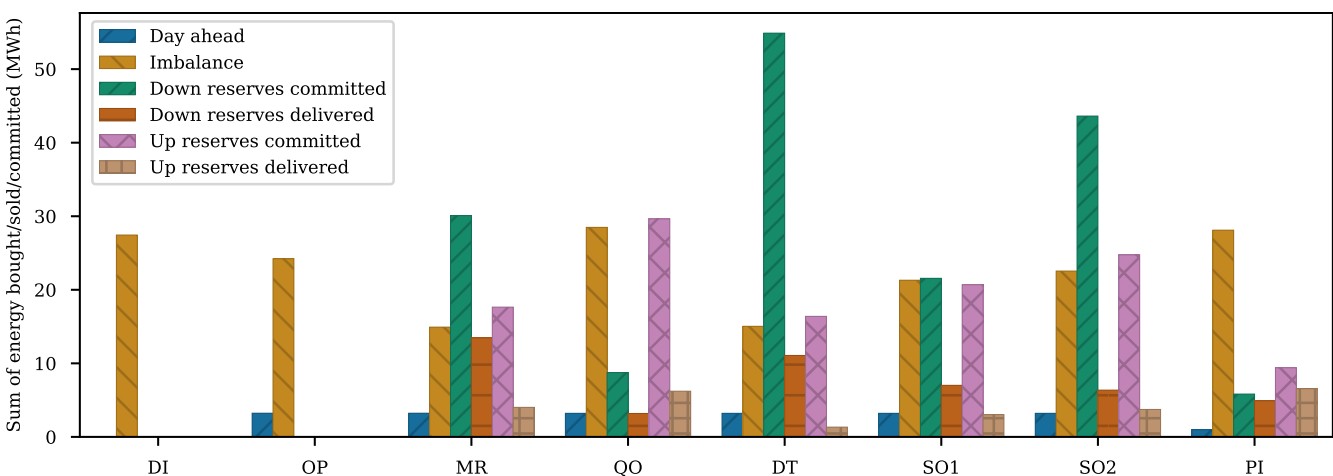

**Figure 4.** Shares of energy bought and sold in DA and imbalance markets, and shares of energy committed and delivered as reserves.

A general observation from the results is that there is a high variance in the results, and as a result, differences between methods are as small as 2% of the total standard deviation. The largest difference is between DI and SO2, with SO2 being 60% cheaper than DI. Having perfect information is on average yet another 85% of the total standard deviation better in comparison to SO2. However, the standard deviation for PI is even higher than the total standard deviation. This means the performance variability is inherent to the problem. This also shows the importance of dealing with uncertainty in this problem.

### 4.2. Controlled Increasing Prediction Quality Case Study

The setup of the case study with controlled increasing prediction quality was similar to the previous case study except for the following point. Instead of generating 25 scenarios per time step, the simulator generated $25q$ scenarios. From these $25q$ scenarios, 25 scenarios with the smallest error were selected. This error measure is defined in such a way that differences between the real and generated scenario at the beginning of a scenario have a higher weight. By changing the parameter $q$, the quality of the forecast can be regulated, with $q = 1$ denoting no information increase, and higher $q$ means higher forecast quality.

Figure 5 and Table 4 show the effect of the increase in information over time. As time progressed, the methods were re-evaluated with more up to date data. The results in Figure 5 show what the end results would have been if no changes in decisions were made from that point. MR, DI, and PI have been left out from this evaluation. MR was left out because the value of $q$ does not influence its decisions as MR ignores the data.

A first glance shows directly that over time the operation costs for all methods decrease. The updated information allows the methods to improve their decisions and this gives better results. Figure 5 shows that the four methods that provide reserves improve more over time than OP. This is because OP only optimizes based on the expected energy price. The other methods provide reserves and learn over time whether reserve bids are accepted and deployed or not, and can update their decisions based on that afterwards.

In the previous case study, the heuristic MR method had a similar performance to more complex methods, such as SO1, even though it does not use any of the provided data. However, with improved data quality, QO, DT, SO1, and SO2 all perform better on average in terms of penalized costs. This shows the importance of analyzing methods with different information quality.

The amounts of exceeded battery capacity and unmet demand also decrease over time. The increased information quality makes only a small difference for the amount of unmet demand at the end of the charging session. For DT, SO1, and SO2, this difference is not statistically significant. Just as with $q = 1$, QO, SO1, and SO2 all scored well below 1% for unmet demand on average. For these methods more than 90% of the time there is no unmet demand or exceeded battery capacity at all. The difference in exceeded battery capacity is bigger and statistically significant for all methods expect for SO1, but SO1 already has almost no exceeded capacity when $q = 1$.

**Table 4.** Results for the controlled increasing forecast quality case. The values shown are the averages across all 950 scenarios for $q = 2 \pm$ the standard deviation, with the average difference between $q = 2$ and $q = 1$ in parentheses (see values in Table 3).

|  | Costs + Penalty (€) | Unmet Demand (%) | Exceeded Capacity (%) |
|---|---|---|---|
| OP | 0.33 ± 0.49 (−0.06) | 0.0 | 0.0 |
| QO | 0.19 ± 0.56 (−0.09) | 0.22 ± 1.17 (+0.14) | 0.06 ± 0.35 (−0.15) |
| DT | 0.16 ± 0.60 (−0.06) | 1.73 ± 2.78 (+0.10) | 0.21 ± 1.18 (−0.12) |
| SO1 | 0.19 ± 0.56 (−0.08) | 0.11 ± 0.63 (−0.00) | 0.02 ± 0.18 (−0.01) |
| SO2 | 0.12 ± 0.63 (−0.07) | 0.26 ± 1.14 (+0.02) | 0.09 ± 0.67 (−0.08) |

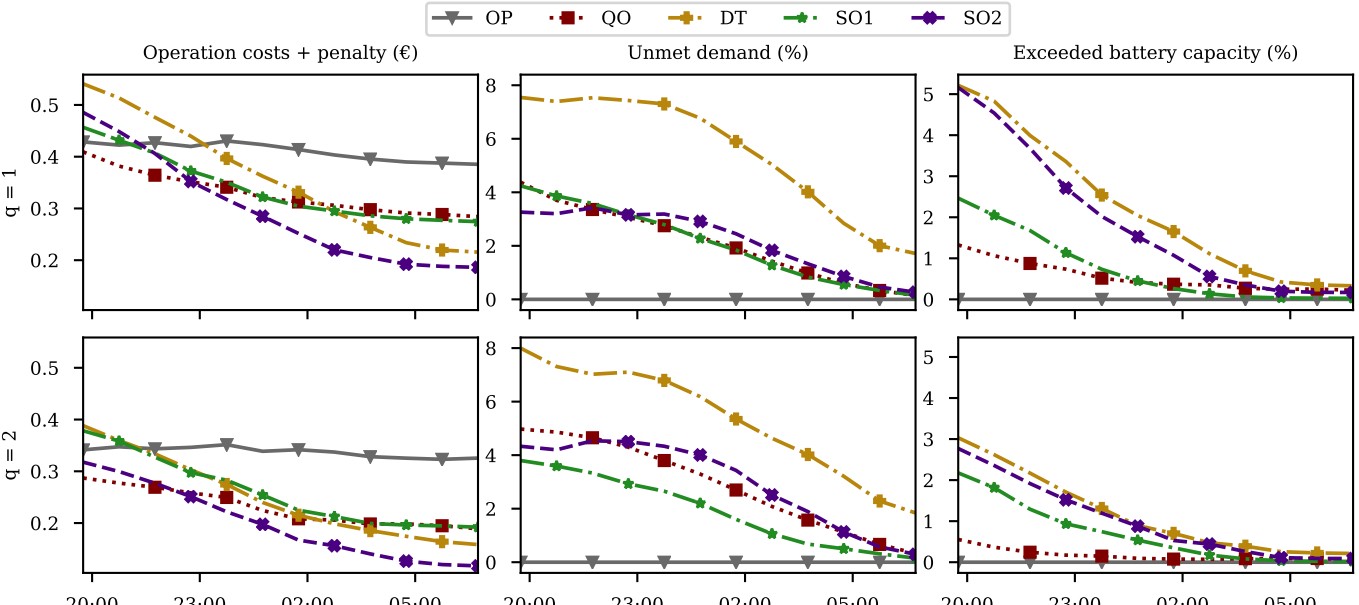

**Figure 5.** Results for the controlled increasing forecast quality case. Forecast quality was regulated by selecting the best 25 scenarios out of 25*q* generated scenarios. Over time, the forecast quality increased. The figure shows the total end results per time step as if the decisions after this time step were not changed. In the case study, new decisions were made every 15 minutes. In the graph, the results are averages for each hour to increase readability.

## 5. Conclusions

The electrification of transportation and the increase of renewable generation can mutually strengthen each other's contributions to the transition to a sustainable energy system. This new energy system requires a so-called smart grid where algorithms that

efficiently match demand to generation play an important role. Large numbers of DR programs and algorithms have been developed for dealing with increasing uncertainty in energy supply, such as by flexibly scheduling the charging of electric vehicles. This paper presents a benchmarking tool for quantitative analysis of such methods, called B-FELSA. B-FELSA is designed to consistently compare flexible load scheduling algorithms under a variety of different market designs and varying information regarding future price and reserve acceptance scenarios.

The two presented case studies illustrate the value of B-FELSA. The results show that a simple analysis of algorithms based on expected values is not good enough, because of the highly volatile nature of imbalance prices and ancillary service usage. In particular, the strategy that uses expected values (DT) had the highest percentages of unmet demand and exceeded battery capacity.

Second, the case studies show that it is hard (very costly) to find schedules with reserves that are robust and meet the minimum energy and capacity requirements in all scenarios. This is of particular importance for stakeholders that want to minimize or quantify risk, or for a TSO that considers using flexible loads for maintaining system stability.

Third, B-FELSA gives insight into how the solution methods improve their decisions over time. The uncertainty of the market behavior makes it difficult to trade based on schedules made the day ahead. The online analysis shows that updating your decisions during the day is necessary to decrease costs and to minimize risk, and that the quality of data should be considered when choosing a scheduling algorithm. It also shows the benefit of using stochastic programming or robust approaches, especially with regard to balancing risk and costs.

This paper focuses on the optimization from the perspective of flexible load consumers, and in particular, owners of electric vehicles or aggregators acting on their behalf. With future work the presented framework should also be able to answer research questions from other stakeholders. A consumer is interested in minimizing costs and risk. Distribution system operators want to prevent congestion and maintain voltage level standards, and are therefore interested in consumer behavior and in providing the right incentives to consumers to maintain stability. Similarly, TSOs are interested in prescribing the balancing capacity available considering the incentives offered by the wholesale market and their imbalance settlements. Furthermore, policymakers want markets that efficiently allocate resources and provide solutions that improve social welfare.

B-FELSA can already answer some of the questions that these stakeholders may have. Still, there are a few important additions that would enlarge the scope of the proposed framework: an implementation of the intraday market, other types of flexible load (in particular, shiftable loads), controllable (distributed) generation (for instance, by micro-CHPs), and different types of local grid constraints are examples. The framework is designed to make such additions straightforward when solution methods for these other devices are given. The case studies also show that the (quality of the) data influences the performance of the algorithms. By adding other data generation method, it will be clearer what part of performance can be explained by the data, and what part can be explained by the solution method. Other future work is to use the insight that B-FELSA provides to make a better online scheduling algorithm.

Regulators must design the right incentive mechanism that enables social welfare of energy system operations. Operators need to make the best daily operations, maintenance, and planning decisions that guarantee a reliable and sustainable service. Users want the best and most affordable service. Moreover, users that can sell to the grid or can offer flexibility for smooth operation may need an incentive and efficient planning algorithm to make their flexibility available. Therefore, there is a high demand for methods to support the diverse set of new decisions faced by the energy sector stakeholders. B-FELSA provides the framework to assess these methods, and hereby has the potential to contribute to a further electrification of transportation and a more sustainable future energy system.

**Author Contributions:** K.v.d.L. designed and implemented the benchmarking framework, performed the experiments, and handled the project as first author. N.R. is the main author of the first draft, except for the discussion of the results. M.M.d.W. supervised the project, acquired the funding for the project, and reviewed and edited the final manuscript. All authors discussed the ideas for the benchmarking method under uncertainty and the results, and approved the publication. All authors have read and agreed to the published version of the manuscript.

**Funding:** This research was funded by the Netherlands Organization for Scientific Research (NWO), as part of the Uncertainty Reduction in Smart Energy Systems program (URSES).

**Conflicts of Interest:** The authors declare no conflict of interest. The funders had no role in the design of the study; in the collection, analyses, or interpretation of data; in the writing of the manuscript, or in the decision to publish the results.

## Abbreviations

The following abbreviations are used in this manuscript:

| | |
|---|---|
| ARMAX | Auto-regressive moving average model with exogenous variables |
| B-FELSA | Framework for benchmarking flexible electric load scheduling algorithms |
| DA | Day-ahead |
| DI | Direct charging model |
| DR | Demand response |
| DSO | Distributed System Operator |
| DT | Deterministic model |
| EV | Electric Vehicle |
| MR | MaxReg heuristic method |
| OP | Optimal price model |
| PI | Perfect information |
| PTU | Program time unit |
| QO | Quantity only model |
| SOC | State-of-charge |
| SO1 | One-stage stochastic programming model |
| SO2 | Two-stage stochastic programming model |
| TSO | Transmission system operator |

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
