# Peer review of "Benchmarking Flexible Electric Loads Scheduling Algorithms"

_energies, doi:10.3390/en14051269_

Round 1

Reviewer 1 Report

.

Author Response

Dear reviewer,

Thank you very much for your thorough review of our manuscript. We’ve gladly used your comments to improve the readability and explain how this work contributes to the literature. Please see our detailed responses to each of the comments below. (Your comments we’ve put in italics and our responses in regular roman font.)

We are convinced that the current draft is a worthy and even stronger contribution to your journal and the field in general, but we’re welcoming any further suggestions of how to improve its presentation.

Best regards,
Koos van der Linden, Natalia Romero and Mathijs de Weerdt

Reviewer 2 Report

This paper considers the problem of balancing the relationship between electricity generation and usage. The authors propose a new benchmarking tool on the scheduling methods for flexible energy consumption. The main novelty is that this benchmarking tool can be used for different objectives and under diverse conditions. Please see the following comments:

  1. In the Introduction part, could the authors use a table/figure to explain the novelty/difference of the proposed benchmarking compared with the existing ones?
  2. Since there are already a large collection of works in this area, the authors are advised to include a related work section to review the recent works. The following works on energy [i-ii] may give the authors some useful info:

[i] Energy-Minimized Multipath Video Transport to Mobile Devices in Heterogeneous Wireless Networks, IEEE Journal on Selected Areas in Communications, vol. 34, no. 5, pp.1160-1178, 2016.

[ii] ``Quality-Aware Energy Optimization in Wireless Video Communication With Multipath TCP, IEEE/ACM Transactions on Networking, vol. 25, no. 5, pp. 2701-2718, 2017. 

  1. In Fig. 1, why the simulation step appears twice in Step 3 and 4? If there is a loop for the simulation, optimization and verification, please clearly explain.
  2. Please adjust Fig. 3 because it is out of the page boundary.
  3. Finally, one major concern of a benchmarking tool is the applicability and feasibility. Could the authors present some examples of evaluating the existing and commonly-used methods?

Author Response

(The authors gave the same response as above.)

Reviewer 3 Report

I can recommend the manuscript for publication in Energies. There are a few minor points I would like to see before publication:

a. the minus sign in the last entry of Table 1 for the costs is probably a typo

b. In the description of the first use case it is mentioned that 'discharging' is not allowed. Does this apply to the use case only, or is V2G in general note considered ?

c. I am color blind and would appreciate if the color-coding in Fig. 3 can be changed to something more accessible. 

Author Response

(The authors gave the same response as above.)

Reviewer 4 Report

In my opinion, the paper is in general interesting and nice to read.

All the following indicated aspects should be clarified and better explained in the manuscript.

General

  1. Differently from what stated at lines 164-166, the heat pumps are comfort based loads (see [3]).
  2. Since EVs are representative of energy based loads, and results of B-FELSA are focused on EVs only, the authors could consider adjusting the title to better reflect the paper contents.

Methodology /  System model

  1. It could be better to insert at the beginning of Section 2 an outline (or moving Fig. 1) about the methodology diagrams flows (how many steps, the aim of each step, the actors involved in each step, etc.); maybe the use of UML or SysML could help authors describing the proposed system view in a more structured fashion.
  2. The Authors should generally describe the system where they apply the proposed methodology, including assumptions and limitations.
    • It seems that the underlying hypothesis of the proposed approach (confirmed by the scenarios analysed in the case study) is that the load is represented by EV only. On the other hand, several recent scientific studies (e.g., https://doi.org/10.1109/TASE.2020.2986269, https://doi.org/10.3390/en12071231, documents that could be referenced in the text), show that scheduling algorithms deal with not only consumers, but also prosumers and energy users equipped with storage capability. Is the proposed method valid to schedule distributed generation and storage features that are today present in energy systems? The authors should comment this point.
    • Recent related works (e.g., https://doi.org/10.1109/MPAE.2007.264850, documents that could be referenced in the text), show that microgrids includes also energy hub and energy router components. Is the proposed method valid for scheduling this kind of components that enables energy sharing? The authors should comment this point.
  3. The description of all (at least the main) the B-FELSA used variables (inputs and outputs) could be listed and grouped in a Table. The authors could also elaborate more on the functionalities indicated in Section 3.2.

Author Response

(The authors gave the same response as above.)

Round 2

Reviewer 2 Report

Thanks to the authors for addressing the comments and including the revisions. The reviewer has no further comments. Please proofread the manuscript and improve the presentation if this paper can be accepted for publication.